# Lympho-SPECT/CT as a Key Tool in the Management of a Patient with Chylous Ascites

**DOI:** 10.3390/biomedicines11020282

**Published:** 2023-01-19

**Authors:** Francesca Iuele, Dino Rubini, Corinna Altini, Paolo Mammucci, Antonio Rosario Pisani

**Affiliations:** Nuclear Medicine Unit, Interdisciplinary Department of Medicine, University of Bari ‘Aldo Moro’, 70124 Bari, Italy

**Keywords:** chylous ascites, robotic laparoscopic prostatectomy, lymphoscintigraphy, SPECT/CT, lymph stasis

## Abstract

Chylous ascites is a rare form of ascites usually associated with cirrhosis, abdominal malignancies, surgeries or infections. We presented a case of chylous ascites after robotic laparoscopic prostatectomy (PLDN-RALP), in which the correct diagnosis was achieved by SPECT/CT lymphoscintigraphy. A 72-year-old male developed chylous ascites after surgery and underwent lymphoscintigraphy with radiolabeled albumin nanocolloids for the supplementary study of the lymph flow and to detect a possible site of leakage. The scintigraphic imaging demonstrated the abdominal effusion and lymph stasis in the left iliac region. The combination of planar imaging with SPECT/CT can resolve the assessment of chylous disorders.

## 1. Introduction

Chylous ascites (CA) is a rare form of ascites, defined as a leakage of triglyceride-rich lymph into the abdominal cavity, with an accumulation of lipid-rich, milk-like peritoneal fluid. Any source of lymph vessel obstruction or leakage can potentially result in CA. The causes of these disorders lead to lymph accumulation in the abdominal cavity and the inability to recycle the accumulated lymph [1,2,3,4].

CA incidence rates are low, and often it is associated with underlying etiologies such as cirrhosis and malignancies. Moreover, non-portal etiologies can cause the exudation of lymph material and can be classified as congenital or acquired. In adults, lymphatic damage can result from trauma, surgeries, or infections [2].

CA has been reported after surgical procedures including retroperitoneal lymphadenectomy and genitourinary surgery, but rarely after laparoscopic procedures such as robotic laparoscopic radical prostatectomy (PLDN-RALP), which have lymphocele formation as the main complication (in almost 50% of patients).

After surgery, CA can occur early (around 1 week) most frequently due to disruption of the lymphatic vessels; the late onset (weeks to months) can be due to adhesions or extrinsic compression of lymphatic vessels. CA represents a critical clinical situation with whole-body mechanical, nutritional and immunological consequences, associated with high morbidity rates and mortality rates of 40–70% [2,5,6,7,8].

CA early diagnosis is crucial and radiological assessment play an important role but neither computed tomography (CT) nor magnetic resonance (MR) is specific for it. Among all lymphatic vessel examination, lymphoscintigraphy is a milestone due to its versatility and the wide spectrum of data that can be analysed [5,9,10].

We present the case of a patient who developed chylous ascites after laparoscopic surgery in whom lympho-SPECT/CT was decisive for the choice of the therapy approach.

## 2. Case Presentation

A 72-year-old male underwent pelvic lymphadenectomy during robotic-assisted laparoscopic radical prostatectomy (PLDN-RALP) for prostate carcinoma eradication (Gleason score 7 = 4 + 3). The patient was discharged on day 6 after surgery thanks to the lack of side effects and the good result of the procedure.

Unfortunately, he was readmitted 7 days later with abdominal distension and dyspnoea. During hospitalization, thorax and abdominal contrast-enhanced CT was performed which confirmed a massive ascites and described the presence of left pleural effusion without the presence of liver diseases (Figure 1).

In order to better assess the post-surgery pelvic region, abdominal MR was performed which did not show evidence of surgical injuries and liver diseases (Figure 2).

Due to copious ascitic fluid, in order to reduce symptoms, he was submitted to paracentesis (1500 mL) and the biochemical analysis of the fluid confirmed the chylous ascites. Four days later, the paracentesis, abdominal distension and respiratory symptoms onset again.

The patient was then submitted to lymphoscintigraphy in order to identify the possible etiology of the relapsing chylous ascites; the day before the scan, the patient observed a diet with high-fat meals in order to decrease the possibility of false-negative results. 

Superficial lymphoscintigraphy was performed, administering 99mTc-labelled human serum albumin nanocolloids (185 MBq) simultaneously on the dorsum of each foot subdermally, about 1–2 cm proximally to the first interdigital space.

Images were acquired with a dual-head gamma camera (Optima NM/CT 640, GE, Waukesha, WI, USA) with low energy high-resolution parallel hole (LEHR) collimators (set 20% window centered on the 140 KeV photopeak).

The patient was positioned supine and a dynamic anterior image of the pelvic region was acquired with a rate of 1 frame per min for 20 min that showed only the left inguinal lymph node, revealing delayed and asymmetric lymphatic drainage from lower limbs. After, anterior whole-body images were acquired at 30 and 60 min and after a high-fat meal at 2 and 3 h after administration (Figure 3). Images at 30 and 60 min confirmed the asymmetric lymphatic drainage with delayed visualization of the right inguinal lymph node (Figure 3a,b). Delayed acquisitions after a high-fat meal, showed diffuse pathological abdominal accumulation of 99mTc-labelled human serum albumin nanocolloids; furthermore, a radioactivity spot was visualized in the left iliac region where all internal iliac lymph nodes should have been removed by surgery (Figure 3c,d).

Abdominal and thorax single photon emission tomography/computed tomography (SPECT/CT) images (128 × 128 matrix, scan step 3, decreased exposure time of 20 s, and each detector rotation angle of 180 degrees) were acquired 3 h after administration. The integrated CT scan was obtained with 110 kV, 75mAs, and pitch 1.3; it allows anatomical localization and image correction attenuation.

Abdominal SPECT/CT images confirmed the radioactive spot in the left iliac region and the diffuse abnormal accumulation of radiotracer in the peritoneum more intense in the left iliac fossa; the CT co-registration images showed that there was no internal iliac lymph node in that site but a dilatation of the lymphatic pathway at the site of surgical interruption, surrounded by a greater intensity of radioactivity in the ascitic effusion. The cause of chylous ascites was then identified as a consequence of the exudation of chyle through the lymphatic dilatation into the peritoneal cavity (Figure 4).

Furthermore, thorax SPECT/CT did not show radioactivity distribution in the left pleural effusion excluding the presence also of chylothorax and demonstrating its reactive aetiology of it (Figure 5).

The patient was then submitted again to paracentesis and treated with conservative therapy: diet (high-protein and low-fat diet with short- and medium-chain triglycerides), and pharmacological measures (to inhibit lymph fluid excretion). The weekly follow-up for the next 3 months revealed a progressive improvement of the clinical condition without CA recurrence and also the disappearance of the reactive pleural effusion.

## 3. Discussion

The diagnosis of chylous ascites is mainly based on the observation of the ascitic effusion which appears as turbid yellow fluid on its chemical, cytological and microbiological analysis [2,11].

CA is not a common event and presents with the symptom of abdominal discomfort (81% of cases) usually 1–14 days after laparoscopic surgery, as well as in our patient that presented abdominal distension and dyspnea 13 days later. The following pain may result from distension of the mesenteric and retroperitoneal serosa as well as in all forms of ascites including neoplastic ones [2,11,12].

The main cause of CA is the presence of chronic liver disease, but in the case of surgery, the possibility of injuries must be considered, even if recent robotic technologies have been implemented and complication rates are even lower [5].

CT is the imaging modality of choice to evaluate intraperitoneal fluid accumulations and may be particularly helpful in the setting of postoperative causes of CA to determine the extent and location of possible injuries [2,11]. In our patient, both abdominal and thoracic CT were performed which demonstrated the presence of a significant amount of CA distributed throughout the peritoneal cavity but also the presence of pleural effusion, on whose etiology was not possible to express considering that the CT density of chylous resembles that of water and is indistinguishable from simple effusion. Furthermore, no alterations attributable to post-surgical damage were identified. Abdominal radiological imaging can include MR, which is not specific to CA but can be useful in identifying intra-abdominal masses, lymph node alterations and liver evaluation [6]. In our patient, there was no imaging evidence of liver, spleen or kidney disease responsible for CA. 

Lymphangiography and lymphoscintigraphy are modalities that can study the lymphatic system detecting vessel damage such as fistulas or leakage from lymphatic channels. Studies have reported a lymphangiography detection rate of 64–86% for leakage sites in patients with CA. Some reports have also suggested a therapeutic role in cases with lymphatic leakage probably thanks to the inflammatory and granulomatous reaction on extravasation induced by contrast agents during the procedure. Otherwise, it is an invasive procedure associated with complications such as infection, pain, intra-alveolar hemorrhage, contrast emboli in the lungs, extravasation of lipiodol into the soft tissue and allergic reactions [2,13].

Lymphoscintigraphy is a nuclear medicine non-invasive procedure, with no adverse effects, no contraindications, and the ability to perform repetitive studies. It allows a functional evaluation of the lymphatic system and has proven valuable in the identification of the sentinel lymph nodes in radio-guided surgery and in the study of limb lymphatic drainage in pre and post-surgical evaluations. Lymphoscintigraphy allows studying both deep and superficial lymphatic vessels by providing a map of the active vessels that drain the injection site, and thus being able to distinguish the altered radiocolloid drainage, the existence of short-circuits, stenosis, bypass or dermal reflux [9,10]. In patients with CA, is useful for evaluating the possible lymphatic alterations underlying it [14].

In our patient, lymphoscintigraphy was used to assess the presence of lymphatic changes after the surgery that involved lymphadenectomy of the iliac lymph nodes. Delayed lymphatic drainage in the right lower limb was found; in subsequent acquisitions, the images showed the presence of radioactivity in the abdominal area confirming the presence of CA and a spot in the left iliac region, which in the two-dimensional planar images could be attributable to an iliac lymph node.

Recent technologies significantly enhanced the diagnostic accuracy of this already valid procedure. Modern gamma cameras allow the acquisition of tomographic and CT co-registration images which improved the diagnostic performance of all scintigraphic examinations.

For the study of lymphatic vessels, lympho-SPECT/CT moved from an ancillary role to being fundamental due to the high accuracy in detecting even minor lesions and the possibility of correlating them anatomically. Our patient is an exemplary case because lympho-SPECT/CT was decisive in characterizing the spot in the iliac area not as a lymph node but as a lymphatic dilatation and in describing the proximal greater peritoneal accumulation of radioactivity.

Images provided by lympho-SPECT/CT then allowed us to identify the CA aetiology in the exudation of chyle through the lymphatic dilatation identified, which is one of the three possible causes of postsurgical CA [1].

Another significant piece of information provided by thorax lympho-SPECT/CT is the absence of radioactivity at the site of the pleural effusion; this allowed us to identify the reactive nature of the effusion, excluding the presence of damage to the connecting lymphatic vessels [15].

The therapy cornerstone for CA regards correcting the underlying cause and applying conservative measures to improve patient comfort, reduce recurrence and optimize outcomes. CA conservative treatment is centered on the use of diuretics and nutritional optimization with a low sodium, low fat, and high protein diet with medium-chain triglycerides in order to reduce the production and flow of lymph. Treatment should be individualized and adjusted for the severity of chylous ascites but refractory cases of CA may require interventions such as embolization, TIPS, peritoneovenous shunting or surgery [2,6]. Thanks to all lympho-SPECT/CT findings were possible to choose conservative therapy based on the correct diet protocol and medical therapy, excluding the need for surgical procedures.

In conclusion, CA after a surgical procedure is a relatively rare complication but still requires a prompt assessment due to the elevated risk of morbidity and mortality. Lympho-SPECT/CT with 99mTc-nanocolloids should be considered in the management with CA thanks to its capability to whole-body assess lymphatic vessels in all three spatial planes. This non-invasive technique allows for identifying the sort and the site of lymphatic drainage in order to promptly begin the correct therapy.

## Figures and Tables

**Figure 1 biomedicines-11-00282-f001:**
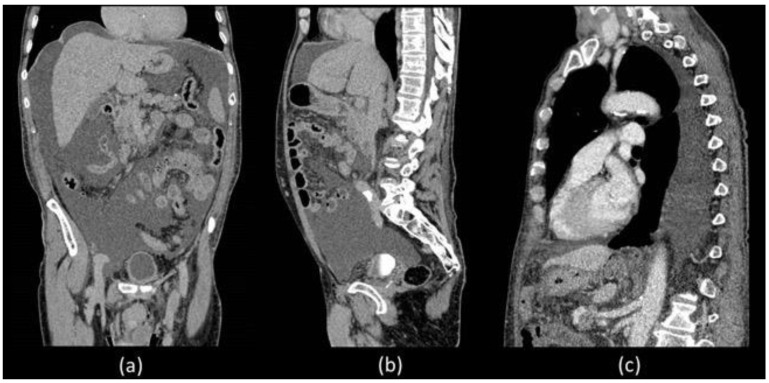
Coronal (**a**) and sagittal (**b**) abdominal CECT showed massive ascites in supra and inframesocolic recesses. Non-hepatic diseases were recorded. (**c**) Sagittal thorax CECT demonstrated left pleural effusion.

**Figure 2 biomedicines-11-00282-f002:**
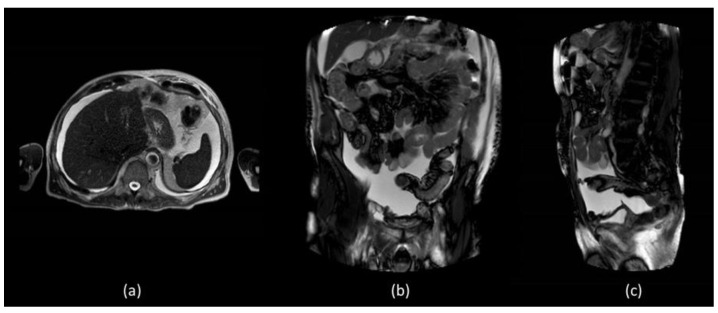
MR T1 weighted (**a**) axial section of the liver did not show diseases but confirmed the presence of massive fluid effusion, visible in the supra and submesocholic space and also in the coronal (**b**) and sagittal images (**c**).

**Figure 3 biomedicines-11-00282-f003:**
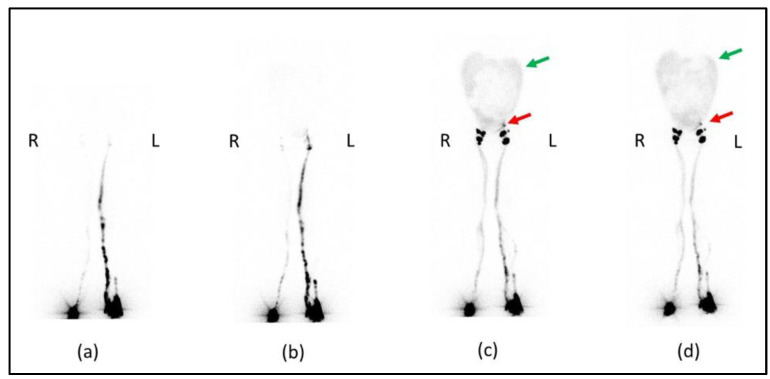
Anterior whole-body lymphoscintigraphy with 99mTc-labelled human serum albumin nanocolloids performed at the 30th minute (**a**) and at the 1st hour (**b**) after administration showed delayed and asymmetric lymphatic drainage of lower limbs. The anterior scans performed after a high-fat meal at the 2nd (**c**) and 3rd hours (**d**) showed bilateral inguinal lymph nodes, one focal spot of radiotracer uptake in the left iliac region (red arrows) and diffuse smooth abnormal abdominal radiotracer uptake (green arrows). R = right side; L = left side.

**Figure 4 biomedicines-11-00282-f004:**
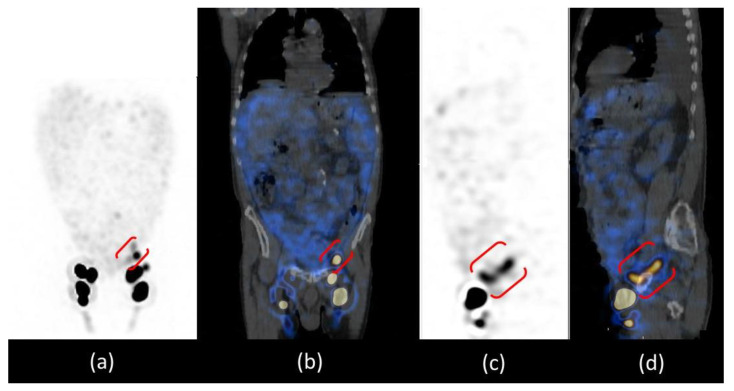
SPET/CT whole body lymphoscintigraphy: MIP (**a**), coronal fused image (**b**), sagittal SPET (**c**) and fused images (**d**). All images confirmed the presence of inguinal lymph nodes and the focal spot of radiotracer uptake in the left iliac region, which in sagittal images is better evidenced as a “whisker” due to the dilatation of the lymphatic pathway at the site of the surgical interruption (red brackets).

**Figure 5 biomedicines-11-00282-f005:**
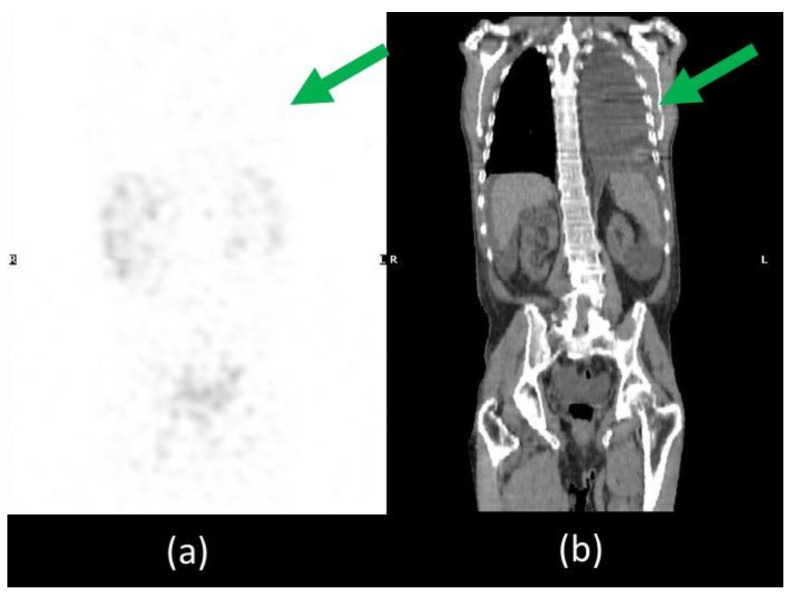
SPET/CT whole body lymphoscintigraphy. MIP image (**a**) did not show any radiotracer uptake in the left pleural effusion visible on CT coronal image (**b**), excluding the presence of chylothorax (green arrow).

## Data Availability

Not applicable.

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
