# Peer review of "Lympho-SPECT/CT as a Key Tool in the Management of a Patient with Chylous Ascites"

_biomedicines, 2023, doi:10.3390/biomedicines11020282_

Round 1

Reviewer 1 Report

The manuscript represents a case report of a relatively rare condition, accompanied by a narrative review of the literature. The images are supportive and suggestive. 

In the introductory part, the authors should also note some considerations about chylothorax. 

The discussion part could be developed even more and few more recent references added.

Author Response

Response to reviewers for“Lympho-SPECT/CT as a key tool in management of a patient with chylous ascites“

Dear Editor, thank you for considering our manuscript for publication.

We followed all the reviewer suggestions, that are all pertinent and directed to improve the manuscript. The comments are not specific for pages and lines, so the modifications are not exactly traceable in the text but they have been all introduced. Here below a point by point response to reviewer.

Reviewer 1:

The manuscript represents a case report of a relatively rare condition, accompanied by a narrative review of the literature. The images are supportive and suggestive. 

In the introductory part, the authors should also note some considerations about chylothorax. 

The discussion part could be developed even more and few more recent references added.

Dear Reviewer, thank you for your support in improving the manuscript.

We followed your suggestion expanding each single part of the introduction, which we have also modified in style, better adapting it to the format of the "case report". Furthermore we added a 2021 reference about robotic laparoscopic prostactectomy and discussion has been modified also following the other reviewer suggestions.

We also submitted the manuscript to a native for editing of English language and style.

Reviewer 2 Report

To authors,

The data is important. I have some advices.

1.     There are too many paragraphs in every section. Especially, “one sentence paragraph” should be avoided in scientific paper. Please combine the present paragraphs. Please look at some other case reports in a first-class journal and “mimic” the style.

2.     Text last: conclusion; templates remain.

3.     References: Templates remain. 

4.     Reference 1,2, 3; templates remain. 

5.     This was due to internal iliac lymphadenectomy, right? The context is unclear. You suggest this in the discussion section. Please state the reason why this occurred. You state the diagnostic details, which is understandable and important. However, please state the reason why this CA occurred. 

6.     You chose “conservative therapy” with success. Please state the criteria to whom surgery is performed and to whom conservative therapy is performed. Since readers are not specialists of CA, please state this. 

7.     Summarizing things, 1) why CA occurred in this patient, 2) not only “localizing” the lesion but also how the present radiological method was useful in deciding the treatment strategy. The context of “pleural effusion” is appropriately described. No need to repeat regarding this part. 

8.     You state that CA is rare in robotic/laparoscopic surgery. This is simply because robotic surgery cases have not been accumulated, right? Then, no need to emphasize “robotic surgery”.  

Author Response

Response to reviewers for“Lympho-SPECT/CT as a key tool in management of a patient with chylous ascites“

Dear Editor, thank you for considering our manuscript for publication.

We followed all the reviewer suggestions, that are all pertinent and directed to improve the manuscript. The comments are not specific for pages and lines, so the modifications are not exactly traceable in the text but they have been all introduced. Here below a point by point response to reviewer.

Reviewer 2:

To authors,

The data is important. I have some advices.

  1. There are too many paragraphs in every section. Especially, “one sentence paragraph” should be avoided in scientific paper. Please combine the present paragraphs. Please look at some other case reports in a first-class journal and “mimic” the style.

Thank you for your suggestion; we have combined more paraghaphs in the revised version.

  1. Text last: conclusion; templates remain.

We apologize for the forgetfulness; the editor corrected them in the "for peer review" version.

  1. References: Templates remain. 

We apologize for the forgetfulness; we have corrected it in the "for peer review" version.

  1. Reference 1,2, 3; templates remain. 

We apologize for the forgetfulness; we have corrected it in the "for peer review" version.

  1. This was due to internal iliac lymphadenectomy, right? The context is unclear. You suggest this in the discussion section. Please state the reason why this occurred. You state the diagnostic details, which is understandable and important. However, please state the reason why this CA occurred. 

The reviewer's question is pertinent, in fact the quoted sentence suggests a causal relationship. We modified the sentence because the execution of lymphadenectomy is an important anamnestic data that indicates the need to study a system that has been handled during a surgical intervention. Lymphoscintigraphy does not investigate the reason for the lymphatic damage but the location and extent of a lymphatic vessels damages. In our case we described that CA occurred because the presence of a lymphatic dilatation in the left iliac region. Furthermore we have accurately described the diagnostic details precisely because the purpose of the manuscript is precisely to describe the information that lympho-SPECT can provide in a non-invasive way.

  1. You chose “conservative therapy” with success. Please state the criteria to whom surgery is performed and to whom conservative therapy is performed. Since readers are not specialists of CA, please state this. 

Thanks to reviewer for this comment; we rewrote the final paragraph of discussion in order to better clarify the choice of the conservative strategy.

  1. Summarizing things, 1) why CA occurred in this patient, 2) not only “localizing” the lesion but also how the present radiological method was useful in deciding the treatment strategy. The context of “pleural effusion” is appropriately described. No need to repeat regarding this part. 

Thank you for these clarifications; we applied them in the discussion section also following previous reviewer comments.

  1. You state that CA is rare in robotic/laparoscopic surgery. This is simply because robotic surgery cases have not been accumulated, right? Then, no need to emphasize “robotic surgery”.  

Thank you, we agree whit this meditation; we applied in the revised manuscript.

Round 2

Reviewer 2 Report

To authors,

The authors faithfully reacted to this reviewer’s advices, of which incorporation into this version has markedly improved the paper quality. The paper clearly illustrates the condition and the usefulness of this diagnostic tool. I believe that this paper may contribute to the further diagnostic/therapeutic improvement of this condition.